# Carbon Storage in Different Compartments in Eucalyptus Stands and Native Cerrado Vegetation

**DOI:** 10.3390/plants12142751

**Published:** 2023-07-24

**Authors:** Fabiana Piontekowski Ribeiro, Alcides Gatto, Alexsandra Duarte de Oliveira, Karina Pulrolnik, Marco Bruno Xavier Valadão, Juliana Baldan Costa Neves Araújo, Arminda Moreira de Carvalho, Eloisa Aparecida Belleza Ferreira

**Affiliations:** 1Embrapa Cerrados, BR-020, km 18, Planaltina 73310-970, DF, Brazilkarina.pulrolnik@embrapa.br (K.P.);; 2Department of Forest Sciences, Universidade de Brasília-UnB, Brasília 70904-970, DF, Brazil; 3Department of Forest Engineering, Centro MultidisciplinarUniversidade Federal do Acre-UFAC, Cruzeiro do Sul 69980-000, AC, Brazil; 4Embrapa Recursos Genéticos e Biotecnologia, Brasília 70770-901, DF, Brazil

**Keywords:** aboveground biomass, Cerradão, underground biomass, GHG mitigation

## Abstract

This study evaluated Carbon (C) storage in different compartments in eucalyptus stands and native Cerrado vegetation. To determine C above ground, an inventory was carried out in the areas where diameter at breast height (DBH), diameter at base height (Db), and total tree height (H) were measured. In the stands, the rigorous cubage was made by the direct method, and in the native vegetation, it was determined by the indirect method through an allometric equation. Roots were collected by direct method using circular monoliths to a depth of 60 cm and determined by the volume of the cylinder. Samples were collected up to 100 cm deep to estimate C stock in the soil. All samples collected directly had C determined using the CHNS elemental analyzer. Gas samples were collected using a manually closed chamber, and the gas concentration was determined by gas chromatography. The results indicate high C storage in the studied areas > 183.99 Mg ha^−1^, could contribute to CO_2_ mitigation > 674.17 Mg ha^−1^. In addition to low emissions (<1 kg ha^−1^ yr^−1^) for the three evaluated areas, with no statistical difference in relation to the Global Warming Potential. Concerning the native cerrado vegetation conversion, the “4-year-old eucalyptus stand” seemed to restore the original soil carbon stocks in the first-meter depth, regardless of some losses that might have occurred right after establishment. Conversely, a significant loss of carbon in the soil was observed due to the alternative setting, where similar natural land was converted into agriculture, mostly soybean, and then, years later, turned into the “6-year-old eucalyptus stand” (28.43 Mg ha^−1^). Under this study, these mixed series of C baselines in landscape transitions have reflected on unlike C dynamics outcomes, whereas at the bottom line, total C stocks were higher in the younger forest (4-year-old stand). Therefore, our finding indicates that we should be thoughtful regarding upscaling carbon emissions and sequestration from small-scale measurements to regional scales

## 1. Introduction

As the World wakes up to the need for efforts towards Climate Change mitigation strategies, the estimates of global greenhouse gas emissions and sinks have become one of the major research priorities of the scientific community. Nevertheless, defining, quantifying, and assessing terrestrial C resources remain challenging, while emissions hit new records yearly. Faced with the ‘Code red’ reality for human-driven global heating [1], natural forests [2,3], Savannas [4,5,6], as well as forest plantations can most easily work as carbon sinks since they fix carbon in their aboveground biomass and also store carbon in roots and soil. Due to land use alteration, environmental services are lost. Payment for ecological benefits can be used to legitimate the option of keeping the original cover [7] or increasing the recovery of degraded areas. Studies suggest that forests play a fundamental role not only in the C cycle but can also contribute to minimizing global warming due to the mitigation of greenhouse gas (GHG) emissions, including nitrous oxide (N_2_O), methane (CH_4_), and carbon dioxide (CO_2_) [8,9,10].

Therefore, the removal of greenhouse gases from the atmosphere by forests and Savannas management can be considered, if not a long-term, at the very least a short-term carbon offsetting. While it will not solve the problem alone, it will play an essential part in the emerging challenges of the Paris Agreement, such as “holding the increase in the global average temperature to well below 2 °C above pre-industrial levels and pursuing efforts to limit the temperature increase to 1.5 °C above pre-industrial levels” [11].

In this 21st conference of the parties (COP21), Brazil has submitted the intended nationally determined contribution (INDC) to reduce its greenhouse gas emissions in 2025 by 37%, compared with 2005 as well as cut off emissions in 2030 by 50%, compared with 2005 [12]. Regarding the land use sector, the Brazilian key policy to tackle climate change has already channeled R$ 17 billion to implement evidence-based mitigation measures within the scope of a Low Carbon Agriculture for Sustainable Development Plan (Plan ABC+). From 2020 to 2030, this public policy comprises the recovery of degraded lands; enhance nitrogen fixation, soil carbon storage, no-till farming, the integration of forest, crops and cattle breeding, agroforestry, and forest planting [12].

Brazil’s commitments also include an effort to apply at least tier 2 in methodologies for land use change, considering four C reservoirs in the forests environment: biomass (aboveground and root biomass), necromass (including dead wood and litter), and soil (soil organic material) [13]. Seen in these terms, the quality of the “Anthropogenic Emissions of Greenhouse Gases National Inventories” depends on the advancement in the use of country-specific Tier 2, which implies replacing the conservative assumptions adopted worldwide (Tier 1) with reliable data that are representative of national and regional reality.

Based on these political settings, obtaining realistic regional metrics on carbon stock measurements and GHG emissions from land use change has been a huge challenge for Brazilian scientists. Brazil has continental dimensions and a diversified economy mostly based on agribusiness products, comprising over 8.5 million square kilometers. This most biodiverse country in the entire world holds equatorial, tropical, and sub-tropical climates, as well as six distinct biomes, namely the Amazon (equatorial rainforest), the Caatinga (semi-arid), the Atlantic Forest (tropical rainforest), the Pantanal (seasonal wetlands), the Pampa (subtropical grasslands) and the Cerrado (savanna) [12].

The Cerrado biome is considered one of the world’s biodiversity hotspots [14], and this region currently has the largest established agricultural area in the country, which led to the loss of 55% of the original altered area of native vegetation [15]. On the other hand, this region faces intense habitat loss [16], accounting for 26% of Brazilian emissions from land-use change [17] 10,688.73 km^2^ yr^−1^ [18].

A Review on Soil Carbon Stocks and Greenhouse Gas Mitigation of Agriculture in the Brazilian Cerrado showed a range of studies conducted on the plot scale when native vegetation in Cerrado was converted to pasture or crops [19]. However, despite the expansion of eucalyptus culture in this Savanna environment [20], scarce information is available concerning the impact of Eucalyptus plantations on carbon stocks and GEE emission [10,21,22,23,24,25,26]. Regarding recently planted forests, Brazil added up to an area of around 10 million hectares, of which 76% were eucalyptus plantations, the world’s most planted hardwood tree [27].

Most research is restricted to aboveground biomass C and soil carbon measurements. There is a gap in estimates of the amount of C in the different Eucalyptus forest compartments, such as roots, and dead biomass in the Cerrado [25,28,29,30]. As for greenhouse gas, only a few sites have been assessed in the Cerrado Biome, and the authors found that CH_4_ removal helped offset greenhouse gas emissions in eucalyptus stands and native Cerrado ecosystems [10,26]. Given the importance of forests in fixing C, considering that in terrestrial ecosystems, vegetation and soil are the leading global sinks of C, information on aboveground and root C is essential to reduce uncertainties in regional metrics about areas of eucalyptus stands in the Cerrado, in addition to feeding national models of C storage and climate change.

Given the hypothesis that C stocks can have different contributions among compartments and studied areas, the following objective was formulated: Quantify, using direct methods, the biomass and C stock by compartments in eucalyptus plantations (*Eucalyptus urophylla* × *Eucalyptus. grandis*) of 4 and 6 years old; Estimate, by indirect methods, the biomass and C stock of the native vegetation of the Cerrado; Estimate the root biomass and C stocks in eucalyptus stands and native vegetation in the Cerrado; Estimate C stocks and assess the rate of C sink in the soil up to 1.0 m depth and; sink Evaluate methane (CH_4_) and nitrous oxide (N_2_O) fluxes from the soil in eucalyptus plantations with different ages and native vegetation in the Cerrado.

## 2. Materials and Methods

### 2.1. Study Area

The study was conducted in the rural area of Paranoá, in the Federal District, Brazil, in areas with two stands of eucalyptus (hybrid *Eucalyptus urophylla* × *Eucalyptus grandis*) of 12 hectare (ha), four years old, clone EAC1528 (A1) and 19 ha, six years old, clone GG100 (A2), and native Cerrado, i.e., Savanna Forest (Cerradão) with 3.29 ha (A3) (Figure 1).

The eucalyptus clonal stand (EAC1528) was established in 2011 at coordinates 15°53′ S, 47°39′ W, and an altitude of 948 m, with spacing 3.5 × 1.7 m. Previously, the site was covered by native Cerrado vegetation. The soil was prepared using a turned-over with a harrow in 15 cm strips and subsoiling in the planting row to a depth of 25 cm.

In December 2009, the eucalyptus clonal stand (GG100) was implemented at coordinates 15°53′ S, 47°38′ W, and an altitude of 946 m, with a 3.5 × 1.7 m spacing. The area’s history was for agricultural use, being used between 2003 and 2005 for the cultivation of soybeans, in 2006 for the cultivation of sorghum, and in the period from 2007 to 2009, again soybeans. Soil tillage consisted of plowing with a plow in 15 cm strips and a furrower in the center of the row to a depth of 25 cm.

In the stands, soil acidity was corrected with the application and incorporation of dolomitic limestone (2.5 t ha^−1^) to a depth of 20 cm, and after two months, 700 kg ha^−1^ of agricultural gypsum was added to the soil surface. The planting fertilization consisted of 200 g seedling^−1^ of NPK (5-25-15). After one year, side dressing was applied with 60 kg ha^−1^ of K_2_O in the form of potassium chloride, 50 kg ha^−1^ of N in the form of urea, and 1 g of boron per plant in the form of borax. In January 2014, another application of 60 kg ha^−1^ of K_2_O was carried out.

The soil of the areas was classified as dystrophic oxisoil (Latossolo Vermelho), and the climate of the region is Aw tropical rainy according to the Köppen classification, with two well-defined seasons: dry (between May and September) and rainy (between October and April). The average annual precipitation observed was 1345.8 mm. Soil chemical parameters for the three environments are presented in Table 1. Clay varied from 58% to 64% for A1, 60% and 72% for A2, and 64% to 69% for A3.

### 2.2. Data Collection and Analysis

#### 2.2.1. Aboveground Biomass of Native Cerrado Vegetation

A forest inventory was conducted to study the floristic composition in the area. The vegetation was sampled in 20 × 50 m georeferenced rectangular plots, and the edge effects were removed. The plots were allocated by an entirely randomized sampling method using the ArcMap program, which is the central application of ArcGIS, version 9.3.

In each sampled plot, all living and dead-standing woody trees with Db (diameter taken at 30 cm from ground level) equal to or greater than 5 cm [31] were botanically identified, and the Db variables and total height were measured. The diameter of each stem was taken with the calipers in two perpendicular directions and calculated the average. The total height (Ht) was measured using a 7.62 m measuring ruler (Crain, model CMR-25). Measurements were taken separately for individuals with multiple stems with Db ≥ 5 cm, branching below 0.30 m from the ground. Density (ind ha^−1^) was calculated without accounting for tree bifurcations, which were only considered for calculating basal area and stem biomass. Individuals were identified botanically, in the field, at the family, genus, and species levels.

The volume, aboveground dry biomass (DB), and aerial C stock were estimated by the non-destructive method using the allometric equation proposed by Rezende et al. [32]:V = ^β^_1_ 0.000109Db^2^ + ^β^_2_ 0.0000154Db^2^Ht(1)
BS = ^β^_0_0.49129 + ^β^_1_0.02912Db^2^Ht(2)
C = ^β^_0_*0.24564 + ^β^_1_*0.01456Db^2^Ht(3)
where: V = individual volume per tree (m^3^), BS = dry biomass (kg·ind^−1^), C = carbon stock (kg·ind^−1^), Db = diameter of the base, taken at 0.30 m from the ground (cm) e Ht = height (m). * Model parameter; (1) Naslund model; (2) and (3) Spurr’s combined variable model.

#### 2.2.2. Aboveground Biomass in Stands

In the delimited plots of 20 × 30 m inside the forest stand of each area, the inventory was carried out, measuring the diameters at breast height (DBH) of all the trees included in the plots to obtain the average DBH, each stem was measured with a caliper, in two perpendicular directions and the average calculated. Total height (Ht) was measured using an electronic clinometer and hypsometer (Haglõf HEC^−2^). Five trees of a medium diameter and representative of the population were selected in each area. Direct measures of rigorous cubing were used for the eucalyptus stand to estimate the biomass. To collect dendrometric data, each tree was felled and separated into different compartments and the shaft was cubed by the Smalian method to obtain its volume.

Nine random samples were used to estimate the litter biomass in each area. For this purpose, a square metallic template with dimensions of 0.5 × 0.5 m (0.25 m^2^) was randomly placed with a minimum distance of five meters between the sampled points. Data were collected in places with similar topographical conditions and at a minimum distance of 20 m from roads and firebreaks to avoid possible edge effects on the litter stock.

The samples from the aerial compartments were weighed on a precision scale to obtain the wet mass and placed to dry in a forced ventilation oven (65 ± 5 °C) until constant biomass was obtained. Subsequently, the samples per tree were sent to the laboratory for the determination of C performed using the CHNS analyzer (Elementary Analyzer of the Macro Vario Cube-Elementary).

#### 2.2.3. Root Biomass

The procedure for collecting roots from native vegetation consisted of excavating five monoliths with a circular shape, with an average diameter of 23 cm and a depth of 60 cm adapted from the Gatto et al. [25] method, placed in the center of each sample plot used in the floristic inventory. A total of 50 monoliths were excavated.

In collecting roots from eucalyptus stands, the collection procedure was similar to native vegetation. A medium CAP tree was selected, per plot, without flaws in the neighborhood, with a well-formed crown, which was felled to determine aerial biomass and collection of roots. Ten circular-shaped monoliths were excavated, with an average diameter of 23 cm to a depth of 60 cm, distributed in four cardinal quadrants (north, south, east, and west), based on the center stump of the eucalyptus tree felled Gatto et al. [25].

In the tree planting line, 50 cm equidistant from the vine, two monoliths were opened, and perpendicular to the planting line (between the rows), three monoliths were opened, also 50 cm equidistant between the points from the tree vine. In separating the roots, the same procedure adopted for native vegetation was followed. The soil collected from each monolith was identified and taken to the laboratory to separate roots and other plant debris shortly after collection by manual process and with the aid of sieves.

Roots separated from the soil, and other plant materials were washed in running water to remove excess soil. Then, the root samples were weighed on a precision scale (0.01 g) to obtain the wet mass and placed to dry in a forced ventilation oven (65 ± 5 °C) until dry biomass was obtained. Subsequently, root samples collected per tree were sent to the laboratory to determine organic C using the elemental analyzer CHNS (Elementary Analyzer of the Macro Vario Cube-Elementary). After obtaining the dry mass, the roots were calculated using the volume of the cylinder: V = π·r^2^·h

#### 2.2.4. Soil Collection and Analysis

Soil density was determined by collecting soil samples in three trenches in each area at various depths: 0–5, 5–10, 10–20, 20–30, 30–40, 40–60, 60–80, and 80–100 cm. To measure the C levels in the soil, 20 samples were taken with a Dutch auger (forming a composite), for intervals up to 30 cm deep and five simple samples, in intervals up to 100 cm. The collected samples were dried in the shade and ground with an agate mortar. The total C in the soil was determined using the elemental analyzer CHNS (Elemental Analyzer of the Macro Vario Cube-Elementary).

Each plot constitutes a set of 24 composite samples, originating from three positions in relation to the tree alignment and eight layers, as previously described. The positions in the plot will correspond to the samples taken: (i) from the center of the tree planting rows, (ii) 1/4 (25%) of the width of the rows, and (iii) in the row. Three simple samples were obtained per depth in each of the three positions to compose a composite sample. The single samples were collected with a Dutch auger, except for a sample from the 1/4 position between the rows and a sample from the planting line, where the single samples were collected in the trench (≈1 m^3^). Undisturbed samples for soil density and other attributes were also obtained from this trench.

Soil density was evaluated by collecting triplicate samples at each depth (in the center of the layer), using three walls of the trench, with the aid of stainless-steel volumetric rings, with an internal wall of 2 mm, 5 cm in diameter and 8 cm in height according to the procedure in the Embrapa soil analysis manual [33]. Subsequently, the samples were dried in an oven with forced air circulation at 110 °C until they reached a constant weight for subsequent calculation of soil density.

The calculation of C stocks was conducted for each soil layer in every study area using the equation provided by Veldkamp [34]:Est C = (C × Ds × e)/10)(4)
where: Est C = C stock at a given depth (Mg ha^−1^), C = total C content at the sampled depth (g kg^−1^), Ds = soil density at the depth (kg dm^−3^), e = thickness of the considered layer (cm).

#### 2.2.5. Measurements of CH_4_ and N_2_O Fluxes

The CH_4_ e N_2_O fluxes were measured from October 2014 to September 2015, and the closed chamber method was used for measurements [35], with a sampling frequency of three times per month. Three plots of 20 × 30 m were randomly delimited in each environment. Four chambers were installed in each plot, two in the eucalyptus rows and two in-between rows, spaced about 10 m apart. In the native vegetation, the four chambers were randomly positioned in each plot, resulting in 36 closed chambers installed in each area.

Each closed chamber consisted of a metal base (0.38 m × 0.58 m) inserted 5 cm into the soil and an upper part of PVC (9.5 cm high) coated with a thermal aluminum blanket, which, together with the metallic base, sealed the space delimited by the chamber, for later gases collection and determination. In the upper part of each chamber, a central hole connected to a rubber hose and a three-way valve made it possible to control the output of gases during sampling. In one of the four chambers, a digital thermometer was attached to monitor the average temperature inside the chambers.

Another digital thermometer (incoterm model) was inserted into the soil to determine the temperature at a depth of 5 cm at the established time for gas collection. Gas sampling started at 9:00 am, representing the average of the daily emission condition [35]. Air samples inside the chambers were collected at 0, 15, and 30 min after closing them. Sixty-milliliter polypropylene syringes with a three-way valve attached, in which 30 mL of gases were collected and transferred to pre-evacuated vials. In addition, the atmospheric air standard was collected in each plot to reference the determination of the gas samples. Before and after sampling, the vials were transported in ice-cooled thermal boxes and stored in a refrigerated environment at 18 °C for measurements.

The CH_4_ e N_2_O concentrations were determined with a gas chromatograph (Trace 1310 GC ultra, Thermo Scientific™ Milan - Italy) equipped with a Porapak Q column at 65 °C, an electron capture detector (ECD), and a flame ionization detector (FID). By integrating the area under the curve, it was possible to obtain the results of each sample’s analysis, indicating the gas concentration changes. The following standards were used: 200 ppb, 600 ppb, 1000 ppb e 1500 ppb de N_2_O e 1000 ppb, 5000 ppb, 10,000 ppb e 50,000 ppb de CH_4_. The CH_4_ e N_2_O fluxes were measured by the linear variation in gas concentration in relation to the incubation time in chambers and calculated by the equation as proposed by Bayer et al. [36]:Flux = δC/δt(V/A)m/Vm;(5)
where in the flux (µg m^−2^ h^−1^); δC/δt is the change in gas concentration (nmol N_2_O e CH_4_ h^−1^) in the chamber in the incubation interval (t); V and A are, respectively, the chamber volume (L) and the soil site covered by the chamber (m^2^); m is the molecular weight of N_2_O e CH_4_ (µg), e Vm is the molar volume at the sampling temperature (L).

#### 2.2.6. Global Warming Potential and Equivalent Carbon

The global warming potential of emission GWP (over a 100-year horizon) expressed in CO_2_eq was calculated by multiplying the accumulated emissions of each gas by its radiative forcing. For this, the gas conversion factor of 34 and 298 CO_2_ kg ha^−1^ was used by Myhre [37] through the following equation:GWP = (CH_4_ × 34) + (N_2_O × 298)(6)
where: GWP is the global warming potential (kg CO_2_ eq ha^−1^ano^−1^), N_2_O and CH_4_ correspond to the accumulated emissions of each gas (kg ha^−1^) and the conversion factors used 34 for CH_4_ and 298 for N_2_O.

### 2.3. Statistical Analyses

The above and belowground C storage, GWP and CO_2_eq were subjected to descriptive statistical analysis and applied to the Shapiro–Wilk normality test, followed by the analysis of variance (ANOVA). The GWP and CO_2_eq had a non-normal distribution, so a nonparametric Kruskal–Wallis test of medians was performed at 5% probability to find possible differences between the areas and the years studied by comparisons. Tukey’s test (*p* < 0.05) was used to compare means among areas that had normality in their compartments.

## 3. Results and Discussion

### 3.1. Aboveground and Belowground Tree Biomass and Carbon Pools

#### 3.1.1. C Stock in the Cerrado Vegetation

In the native vegetation of the Cerrado (A3), i.e., Savanna Forest (Cerradão), 84 species and 41 botanical families were found, with a density of 1.858 ind ha^−1^, with Db ≥ 5 cm. The diameter of the individuals sampled ranged from 5 to 60 cm, and the height from 0.30 to 13.5 m with an average of 4.31 m. Table 2 presents the dendrometric characteristics, where the total basal area, volume, woody biomass, and carbon stock of the sampled trees were: 21.0197 m^2^ ha^−1^, 50.6475 m^3^, 44201.59 kg ha^−1^, 22,100.78 kg ha^−1^, respectively. Dead individuals represented 15.3% of the carbon in the studied vegetation. The Brazilian Cerrado is a heterogeneous ecosystem due to the different phytophysiognomies, which results in a considerable variation in biomass and carbon [38,39].

Native species have varying capacities to store biomass and carbon, for example, *Virola sebifera* and *Qualea grandiflora* respond with 132.13 and 2879.11 kg ha^−1^, respectively (Table 2). The number of individuals, size, and wood density also explain this distinct pattern [40]. *Qualea grandiflora* (161 trees ha^−1^) had the highest carbon contribution, about 17.88 kg ha^−1^ of C per tree, however, proportional to the number of individuals, *Q. parviflora* (24 trees ha^−1^) was the one that most stocked C per individual 77.88 kg ha^−1^.

#### 3.1.2. Carbon Stock of Eucalyptus Stands

The mean of the dendrometric parameters collected by forest inventory in the eucalyptus stand was diameter at breast height (DBH) and height (h): 22.6 cm, 14.6 m to A1, and 28.5 cm, 16.5 m to A2, respectively. The area basal and volume sum was: 23.93 m^2^ ha^−1^, 229.51 m^3^ ha^−1^ to A1, and 30.69 m^2^ ha^−1^, 341.61 m^3^ ha^−1^ to A2, respectively. For the biomass and carbon accumulated in the trees (aerial part), it was verified that the wood presented the highest contribution for A1 (81.36%) and A2 (88.46%) Table 3 followed by the bark compartment. Wood presents the highest contribution of biomass and C in eucalyptus stands [25,41,42]. The other compartments, such as bark, leaves, and branches, may vary according to the characteristics of the study area.

Biomass stock and C storage increased significantly with stand age. Zhou et al. [43] also observed this pattern and found a considerable increase in C storage over the years, with a reduction in biomass increment between 13 and 21 years of the forest stand. Kumar et al. [44] report that eucalyptus stands have a rapid increase up to 6 years.

Rapid accumulation of C stocks in the biomass of eucalyptus stands has been previously observed with estimated accumulation rates of 8.3–12.8 Mg C ha^−1^ yr^−1^ in aboveground components [42,45]. For Cerrado edaphoclimatic patterns, Gatto et al. [42] state that eucalyptus plantations show variation in biomass and C stocks in stands with the same age and spacing. Thus, C stocks for eucalyptus spacing 3 × 2 m at 4 years ranged from 44.28 to 64.69 Mg ha^−1^ and at 6 years from 71.21 a 104.42 Mg ha^−1^.

Dense forest stands, with a higher number of trees per hectare, produced a higher amount of stem biomass and, therefore, a higher stock of stored carbon. This highlights the importance of the number of trees per hectare in determining the total biomass production and, consequently, the removal and storage of atmospheric carbon [46]. Thus, the amount of biomass and C availability depends on a set of local factors, including the age of the stand [43], the handling of forest residues [25], and management and chemical and physical properties of soils [47,48], among other factors.

### 3.2. Soil Carbon

The distribution of C content in the eight soil layers showed significant differences (*p* > 0.05) (Figure 2). For the native Cerrado vegetation (A3), the highest C content was observed in the 0–5 cm, 40–60 cm and 60–80 cm layers (Figure 2a). Evaluating by area, it was observed that the total C content of the soil decreases exponentially with depth. The high content of C in the surface layer of the soil is due to the input of organic material and slow decomposition in planted and natural forests, rich in lignin (10–13%) and a high C:N ratio 76–85 [49,50] factors that increase litter decomposition time and increase C levels in the soil. This material comes from the fall of leaves, branches, and bark of trees. The density of fine roots also contributes to increasing C levels even in soils with low natural fertility, such as Cerrado vegetation [51].

Despite the low fertility, high concentrations of exchangeable aluminum, and high acidity rates in the biome [52,53], eucalyptus stands have been able to expand due to the adaptability of this exotic species to the edaphoclimatic conditions in the Cerrado domain [54,55].

Higher C stocks were observed in the soils of the areas studied in the 0–30 cm layer (Figure 2b). Of the total C stock in the 0–100 cm layer of soil, 48, 50, and 52% of the C stock for areas A1, A2, and A3 are present in this superficial layer (0–30 cm), respectively (Figure 2b). Younger eucalyptus (A1) and native vegetation (A3) had the highest C stocks for all layers studied. This means that after 4 years of eucalyptus implantation, there was no reduction of C in the soil. While the A2 area, with a history of agricultural use for the cultivation of soy and sorghum between 2003 and 2009, contributed to lower values of C in the soil. The higher carbon storage in the upper soil horizons in eucalyptus plantations in the Cerrado is consistent with above and near-surface biomass inputs to soil organic matter [56]. The best alternatives to reduce carbon losses in an ecosystem are maintaining the natural forest cover or recovering the soil through sustainable soil management [57], and adhering to low environmental impact systems.

In the present study, the soil organic carbon stocks under Cerradão (167 Mg ha^−1^) and 4 years old Eucalyptus stands (174 Mg ha^−1^) were similar in the same depth classes and down to 1 m in an oxisol also in Cerrado, as reported from Pinheiro et al. [58]. In Brazil, Cook et al. [48] characterized soil carbon stocks and change over two decades in 306 operational *Eucalyptus* stands across a 1200 km gradient (ranging from 18 to 26 years or approximately three to four rotations) showed a slight decrease (−0.22 ± 0.05 Mg ha^−1^ yr^−1^, *p* < 0.0001; 0–30 cm) or no net change in soil carbon stocks, and vary with land-use history and soil properties, precipitation, but weaker associations with soil order and mean annual temperature.

The impact of Cerrado vegetation clearing, burning of residues, and plowing operations have also been reported by Ferreira et al. [59] as major drivers of soil carbon decline during the first years after conversion to crop fields, when a significant proportion of the soil carbon that was physically protected within stable soil aggregates abruptly led to an oxidative condition, resulting in mineralization and the loss as CO_2_.

### 3.3. Carbon Equivalent (C eq) and Global Warming Potential

The results were not influenced by the age of the eucalyptus stands, nor by the replacement of native vegetation by Eucalyptus, considering the time interval since the beginning of the evaluation. During the study period, the accumulated flows were not influenced by the age of the eucalyptus stands or by the replacement of native vegetation by eucalyptus, the variations ranged from 0,33 to 0,85 kg ha ^−1^ year ^−1^. The contributions of the N_2_O fluxes of the Global Warming Potential (GWP) partially ranged from 98 to 271 kg CO_2_ eq ha^−1^ yr ^−1^ for A1, A2, and A3. Considering the annual GWP, the CH_4_ acted as a sink in all studied areas (Table 4) ranged from −21 to −63 kg CO_2_ eq ha^−1^, the contributions of the CH_4_ and N_2_O fluxes of the GWP were 64.89 ± 84, 237.98 ± 119 and 76.61 ± 44 kg CO_2_ eq ha^−1^ for A1, A2, and A3, respectively.

Soils under planted forests and native ecosystems have shown low N_2_O fluxes, which can be attributed to the system’s physical, chemical, and biological characteristics [60,61]. The research focused on GHG emissions in the Cerrado found relationships between emissions and environmental/management variables, but overall N_2_O emissions were low (>2.84 kg ha^−1^) for conservationist or crop-livestock-forestry integrated systems [61,62,63].

The environments studied functioned as sinks for CH_4_ fluxes in the soil-atmosphere system, which caused a reduction in the sum of GWP. The influxes of CH_4_ in the Eucalyptus stand and native Cerrado vegetation are in agreement with reports of emissions from native on non-native forests [64,65,66,67] that indicate CH_4_ uptake from the atmosphere since a higher amount was consumed by the environment than the effectively produced quantity [68,69]. Therefore, this study confirmed that aerated soils can act as a sink or source of CH_4_, generally dominating the methanotrophic metabolism over methanogenesis, which may continue to occur at small anaerobic sites [70,71,72].

### 3.4. Carbon Storage

The C stocks in the soil-plant system showed different contributions in the system in the areas studied, the C stock in the aerial part of the trees was 27.45%, 37.78%, and 12.01%, and in the roots up to 60 cm, it was 2.16%, 0.88% and 1.68% for A1, A2, and A3, respectively. The soil had the highest contribution to total C with 68.17%, 58.38%, and 84.07% for A1, A2, and A3, respectively, functioning as a C pool at all ages of eucalyptus stands [42,43]. However, the soil and native vegetation of the Cerrado are in equilibrium through a specific ratio of input and output of carbon [73].

Comparing the studied areas, A2 had the highest C stock in the aerial part in relation to the others, and A1 had the highest C stock in the roots (Table 5; *p* = 0.05). The lower amount of surface roots in the oldest stand (A2) may be related to the proportion of roots that increase in depth as the age of the stand increases [74]. Litter contributed more than roots to surface C (Table 4) due to its diversity of organic compounds with different chemical complexities, which act as raw material in forming and maintaining soil organic carbon [75]. Eucalyptus stands have good litter production capacity, capable of promoting the entry of organic matter and nutrients into the soil [76,77]. The total C was higher for A1 with a 4 year old eucalyptus stand (Table 4; *p* = 0.05), with a positive balance for the three evaluated areas.

The total annual CO_2_ of a forest ecosystem is positive, discounting the losses by respiration and mortality of plant tissues [3]. However, C is still maintained in long-term tissues and in the soil, validating the importance of native on non-native forests that function as a C sink (>674.17 ± 59.8 Mg ha^−1^ CO_2_eq) and that mitigate climate change, showing their relevance for local sustainability and for the generation of carbon credit trading.

## 4. Conclusions

As first conclusion, the ttotal aboveground C stored in the stands were of 62.1 Mg ha^−1^ for 4-year-old eucalyptus and 81.7 Mg ha^−1^ for 6-year-old eucalyptus. As for the native vegetation, species revelaed wide heterogeneity in terms of C storage, where Qualea grandiflora presented the highest contribution (13%) of the total 22.1 Mg ha^−1^. The roots had a lower contribution to C storage, ranging from 1.9 to 4.9 Mg ha^−1^. However, this contribution is related to a depth of 60 cm, so that it is srongly recommended that the roots biomass data should be collected at least up to one meter deep in future studies. Soil C made the most significant contribution by fixing 58–84% CO_2_ of total C stored. In all areas, N_2_O and CH_4_ emissions were <1 kg ha^−1^ yr^−1^. The soil acted as a continuous CH_4_ sink, showing absorption at rates between −63.09 and −21.53 kg CO_2_ eq ha^−1^ yr^−1^. Concerning the native cerrado vegetation conversion, the “4-year-old eucalyptus stand” seemed to restore the original soil carbon stocks in the first-meter depth, regardless of some losses that might have occurred right after establishment. Conversely, a significant loss of carbon in the soil was observed due to the alternative setting, where similar natural land was converted into agriculture, mostly soybean, and then years later, turned into the “6-year-old eucalyptus stand” (18%; 28.43 Mg ha^−1^). Under this study, these mixed series of C baselines in landscape transitions have reflected on unlike C dynamics outcomes, whereas at the bottom line, total C stocks were higher in the younger forest (4-year-old stand). Therefore, our finding indicates that we should be thoughtful upscaling carbon emissions and sequestration from small-scale measurements to regional scales. Moreover, it is also vitally important to attempt for uncertainties of biogeochemical cycles connected to the carbon and nitrogen cycle. Considering the CHG fluxes, we found indications of a range of CH_4_ GHG removals (12 to 50% of N_2_O emissions), reducing the net annual emissions of these forest systems in different proportions. For this reason, public policy approaches should rigorously quantify, besides uncertainties in metrics of low-carbon transition pathways, the ecosystem-level C stock that includes biodiversity, water cycles, and well-integrated landscape planning approaches before encouraging management practices that maximize carbon sequestration by promoting forest cover as well.

## Figures and Tables

**Figure 1 plants-12-02751-f001:**
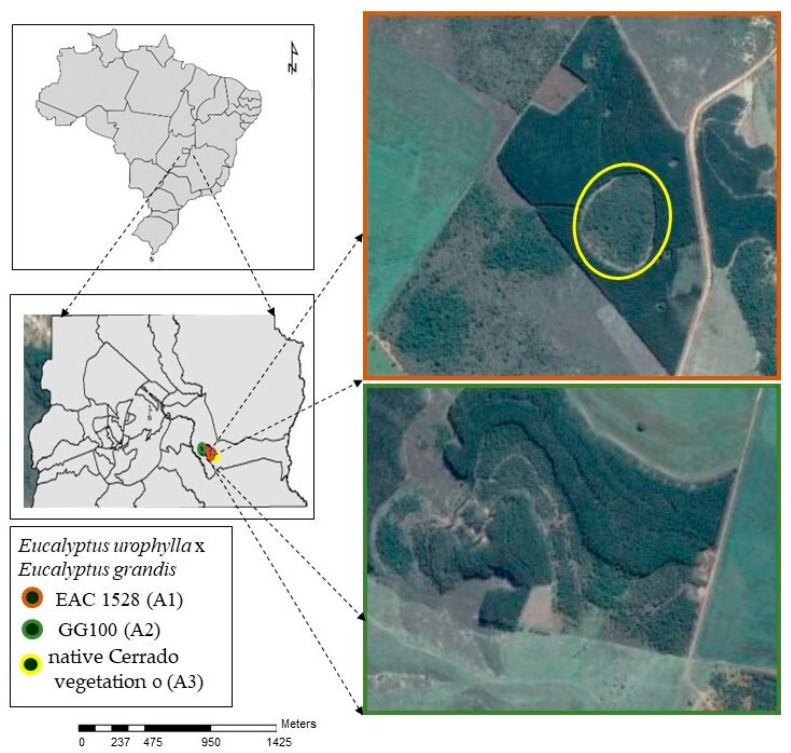
Location of the study area, rural area, Paranoá, in the Federal District, Brazil.

**Figure 2 plants-12-02751-f002:**
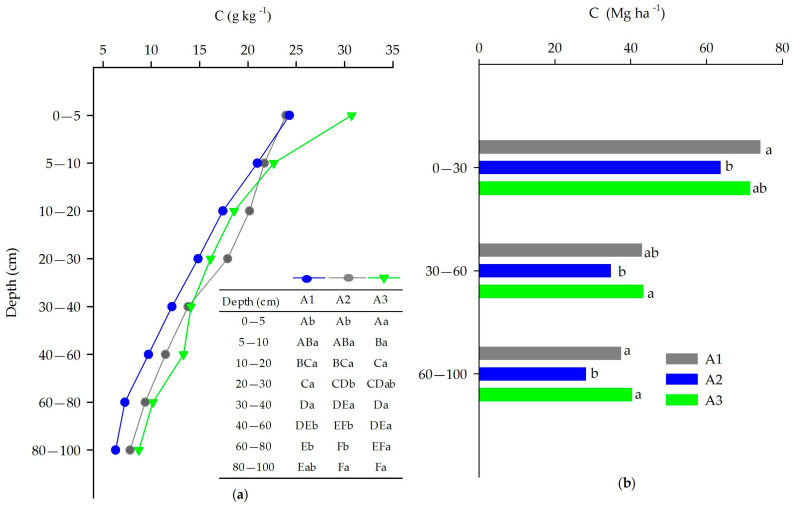
C contents (g kg^−1^) in soil layers up to 100 cm (**a**) and C stocks (Mg ha^−1^) in soil layers up to 0–30, 30–60 and 60–100 cm (**b**) for *Eucalyptus urophylla* × *Eucalyptus grandis* stands with 4 years old clones EAC 1528 (A1) and 6 years old GG100 (A2) and native Cerrado vegetation (A3), in the Federal District, Brazil. Uppercase letters difference between depths and lowercase letters difference between areas, by the Tukey test (*p* < 0.05).

**Table 1 plants-12-02751-t001:** Chemical attributes of soil samples under *Eucalyptus urophylla* × *Eucalyptus grandis* stands with 4 years old clones EAC 1528 (A1) and 6 years old GG100 (A2) and native Cerrado vegetation (A3) in the Federal District, Brazil.

Areas	Depth (cm)	Al^3+^	Ca	Mg ^(1)^	pH H_2_O	H + Al	OM ^(2)^	Mn	K	P	Zn	Fe	B	Density
		---- cmol_c_ dm^−3^---				mg dm^−3^	dag kg^−1^	------------------ mg dm^−3^ ----------------------						g cm^3^
A1	0–5	2.2	0.2	0.6	5.1	9.5	3.8	12.2	230.7	1.8	0.6	142.3	0.5	1.14
A1	5–10	2.3	0.1	0.3	5.1	8.6	3.2	9.1	185.3	1.3	0.4	141.0	0.5	1.24
A1	10–20	2.3	0.0	0.2	5.1	8.0	2.9	6.2	158.7	0.8	0.3	122.0	0.3	1.26
A1	20–30	2.1	0.1	0.2	5.0	7.2	2.6	3.8	109.3	0.5	0.2	59.7	0.3	1.21
A1	30–40	2.1	0.0	0.1	5.1	7.1	2.3	4.9	112.7	0.2	0.3	82.3	0.3	1.19
A1	40–60	1.9	0.0	0.1	5.1	6.2	1.9	4.2	93.3	0.1	0.2	63.0	0.3	1.16
A1	60–80	1.6	0.0	0.1	5.1	5.2	1.1	3.3	83.3	0.0	0.2	55.0	0.2	1.09
A1	80–100	1.4	0.0	0.1	5.1	4.8	1.2	7.3	76.7	0.0	0.2	38.7	0.2	1.08
A2	0–5	2.0	0.8	0.6	5.0	8.2	3.3	12.4	157.3	1.5	0.3	102.0	0.5	1.2
A2	5–10	2.1	0.7	0.5	4.9	8.3	2.9	10.7	138.0	1.4	0.3	101.7	0.5	1.2
A2	10–20	2.3	0.5	0.4	4.8	8.0	2.5	6.4	111.3	1.2	0.3	61.3	0.5	1.2
A2	20–30	2.4	0.1	0.2	4.9	8.1	2.2	6.3	121.3	1.0	0.3	106.0	0.5	1.1
A2	30–40	2.3	0.1	0.1	4.8	7.2	2.0	3.3	90.0	0.8	2.0	94.3	0.3	1.1
A2	40–60	2.1	0.1	0.1	5.0	6.4	1.7	3.1	80.0	0.4	0.2	63.3	0.2	1.1
A2	60–80	1.8	0.0	0.1	5.1	5.5	1.2	3.2	74.7	0.2	0.1	45.0	0.2	1.0
A2	80–100	1.5	0.1	0.1	5.3	4.6	1.1	2.8	70.7	0.2	0.1	38.7	0.2	1.0
A3	0–5	0.6	2.5	0.9	5.4	7.0	3.0	12.7	347.3	5.3	0.6	68.0	1.2	1.11
A3	5–10	1.0	1.7	0.6	5.3	7.1	2.6	9.4	134.7	3.5	1.3	85.3	0.9	1.17
A3	10–20	1.5	0.7	0.3	5.0	6.8	2.3	4.7	148.0	2.1	0.4	87.0	0.8	1.15
A3	20–30	1.8	0.5	0.2	4.9	6.3	2.0	3.1	75.3	1.1	0.2	80.0	0.9	1.22
A3	30–40	1.5	0.6	0.2	5.0	5.6	1.6	3.0	62.7	0.7	0.3	71.0	1.1	1.12
A3	40–60	1.2	0.3	0.1	5.1	5.0	1.2	2.1	47.3	0.4	0.2	51.3	1.6	1.03
A3	60–80	0.8	0.2	0.1	5.2	4.1	0.9	1.8	44.0	0.4	0.1	42.7	0.9	1.06
A3	80–100	0.7	0.2	0.1	4.9	4.0	0.8	1.4	43.3	0.4	0.3	43.0	0.6	1.07

^(1)^ Magnesium; ^(2)^ Organic matter.

**Table 2 plants-12-02751-t002:** Averages of total basal area, volume, woody biomass and carbon stock of trees sampled in native vegetation.

Species	Basal Area	Volume	Biomass	Carbon
	m^2^ ha^−1^	m^3^	kg ha^−1^	kg ha^−1^
*Qualea grandiflora*	2.59	6.42	5758.24	2879.12
*Miconia pohliana*	3.12	7.08	5655.73	2827.87
*Qualea parviflora*	1.05	3.29	3701.77	1850.89
*Xylopia aromatica*	1.41	3.52	3233.56	1616.78
*Emmotum nitens*	0.70	2.13	2342.85	1171.43
*Curatella americana*	0.62	1.57	1434.87	717.43
*Amaioua guianensis*	0.51	1.41	1411.61	705.80
*Miconia albicans*	0.83	1.73	1268.01	634.00
*Terminalia agentea*	0.42	1.19	1241.02	620.51
*Astronium fraxinifolium*	0.48	1.25	1180.94	590.47
*Eriotheca pubescens*	0.33	1.01	1098.38	549.19
*Xylopia brasiliense*	0.37	0.89	791.30	395.65
*Alibertia edulis*	0.42	0.94	777.53	388.76
*Kielmeyera coriacea*	0.54	1.10	753.74	376.87
*Rudgea viburnoides*	0.30	0.69	578.97	289.49
*Simarouba versicolor*	0.19	0.53	541.49	270.74
*Schefflera macrocarpa*	0.18	0.48	491.72	245.86
*Eugenia dysenterica*	0.20	0.46	378.78	189.39
*Ouratea grandiflora*	0.14	0.35	298.20	149.10
*Virola sebifera*	0.09	0.25	264.26	132.13
Other species in total	2.41	5.37	4223.35	2111.67
Dead trees	4.07	8.93	6749.48	3374.74
Vine	0.03	0.06	25.79	12.89
Total	21.02	50.65	44,201.59	22,100.78

**Table 3 plants-12-02751-t003:** Average estimates of dry biomass and Carbon (Mg ha^−1^) stocks of *Eucalyptus urophylla* × *Eucalyptus grandis* stands with 4 years old EAC 1528 (A1) and 6 years old GG100 (A2) clones in the District Federal, Brazil.

Compartment	A1			A2		
	Biomass	Carbon		Biomass	Carbon	
	---Mg ha^−1^ ---		(%)	---Mg ha^−1^---		(%)
Leaves	5.3	2.4	3.76	2.3	1.0	1.21
Branches	5.3	2.4	3.76	3.4	1.5	1.79
Wood	114.8	50.3	81.36	167.8	72.5	88.46
Bark	15.6	7.1	11.06	16.2	6.8	8.54
Aboveground total	141.1	62.1	100	189.7	81.7	100

**Table 4 plants-12-02751-t004:** Cumulative fluxes and contribution of the CH_4_ and N_2_O gas to the Global Warming Potential (GWP) of N_2_O and CH_4_ for *Eucalyptus urophylla* × *Eucalyptus grandis* stands with 4 years old clones EAC 1528 (A1) and 6 years old GG100 (A2) and native Cerrado vegetation (A3), in the Federal District, Brazil. Uppercase and lowercase letters difference between areas for GWP by the Tukey test (*p* < 0.05).

		N_2_O	
	kg ha^−1^ yr^−1^	GWP100 ^1^	kg CO_2_ eq ha^−1^ yr ^−1^
A1	0.43 (±0.23) a	298	127.82 (±77) a
A2	0.85 (±0.45) a	298	271.30 (±116) a
A3	0.33 (±0.20) a	298	98.15 (±59) a
		CH_4_	
A1	−1.85 (±1.36) a	34	−63.09 (±46) a
A2	−0.98 (±0.91) a	34	−33.31 (±31) a
A3	−0.63 (±0.53) a	34	−21.53 (±18) a

^1^ with inclusion of climate–carbon feedbacks, over a 100-year horizon. Source: Myhre [37].

**Table 5 plants-12-02751-t005:** Total estimates of C storage and CO_2_ eq (Mg ha^−1^) in *Eucalyptus urophylla* × *Eucalyptus grandis* stands with 4 years old clones EAC 1528 (A1) and 6 years old GG100 (A2) and native Cerrado vegetation (A3), in the District Federal, Brazil.

Compartment	A1	A2	A3
	--- Mg ha^−1^ ---		
Aerial part	(62.1 ± 3.6) b	(81.7 ± 22.2) a	(22.1 ± 0.02) c
Roots 0–60 cm	(4.9 ± 2.1) a	(1.9 ± 0.84) b	(3.1 ± 0.95) ab
Litter	(5.0 ± 0.23) ab	(6.4 ± 0.2) a	(4.1 ± 0.28) b
Plant total	(72 ± 6.0) ab	(90 ± 23.3) a	(29.3 ± 1.3) c
Soil (0–100 cm)	(154.23 ± 15.64) a	(126.26 ± 13.35) b	(154.69 ± 15.05) a
Soil/plant total	(226.23 ± 21.6) a	(216.26 ± 36.7) b	(183.99 ± 16.3) c
Stored CO_2_eq	(828.94 ± 79.3) a	(792.41 ± 134.3) b	(674.17 ± 59.8) c
Emission CO_2_eq	(0.06 ± 0.08) a	(0.24 ± 0.11) a	(0.07 ± 0.04) a

Uppercase and lowercase letters difference between areas by the Tukey test (*p* < 0.05).

## Data Availability

Data supporting reported results can be acquired by contacting the corresponding author.

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
