# Peer review of "Carbon Storage in Different Compartments in Eucalyptus Stands and Native Cerrado Vegetation"

_plants, 2023, doi:10.3390/plants12142751_

Round 1
Reviewer 1 Report
Comments and suggestions can be found in the attachment.

English is fine. Minor editing may be needed.
Author Response
Dear
We thank the reviewers for their generous and thoughtful comments on our submitted manuscript. We have made all the efforts to address all the reviewers’ concerns. We have included here the reviewers’ comments and how we have addressed each point, in italics the reviewers’ comments and in color green our response.
- -Reviewer 1 –
1.Abstract part should need to be modified. The authors should write a short description of the methodologies used in this study. The coefficient values of the allometric equation should not be put into the abstract part. Instead, the authors should give specific description of allometric equations for determining the AGB (aboveground biomass) and root biomass with mentioning proper references for Eucalyptus stands and native Cerrado vegetation in the methodology part.
- In the Abstract part, the findings (results) and conclusions of the study were not well written. There is also no clear message for the future use of this study
This study evaluated Carbon (C) storage in different compartments in eucalyptus stands and native Cerrado vegetation. To determine C above ground, an inventory was carried out in the areas where diameter at breast height (DBH), diameter at base height (Db), and total tree height (H) were measured. In the stands, the rigorous cubage was made by the direct method, and in the native vegetation, it was determined by the indirect method through an allometric equation. Roots were collected by direct method using circular monoliths to a depth of 60 cm and determined by the volume of the cylinder. Samples were collected up to 100 cm deep to estimate C in the soil. All samples collected directly had C determined using the CHNS elemental analyzer. Gas samples were collected using a manually closed chamber, and the gas concentration was determined by gas chromatography. The results indicate high C storage in the studied areas. In order to determine C above ground, an inventory was carried out in the areas where diameter at breast height (DBH), diameter at base height (Db), and total tree height (H) were measured. In the stands, the rigorous cubage was made by the direct method, and in the native vegetation, it was determined by the indirect method through an allometric equation. Roots were collected by direct method using circular monoliths to a depth of 60 cm and determined by the volume of the cylinder. Samples were collected up to 100 cm deep to estimate C in the soil. All samples collected directly had C determined using the CHNS elemental analyzer. Gas samples were collected using a manually closed chamber, and the gas concentration was determined by gas chromatography. The results indicate high C storage in the studied areas>183.99 Mg ha-1, generating CO2 mitigation > 674.17 Mg ha-1. In addition to low emissions (< 1 kg ha-1 yr-1) for the three evaluated areas, with no statistical difference in relation to the Global Warming Potential. Currently, there is a shortage of research on the total carbon estimates for eucalyptus stands and native vegetation in the Cerrado region. This study aims to provide data to improve models that reduce uncertainties in forest compensation through carbon.
- In the Introduction part, there is a lack of the study’s aim (s) of which the study is targeting to address. Only research hypotheses are not enough. The Introduction part needs to highlight the relevance of standing biomass and soil carbon on carbon offsetting to reduce GHG emissions
As the World wakes up to the need for efforts towards Climate Change mitigation strategies, the estimates of global greenhouse gases emissions and sinks have become one of the major research priorities of the scientific community. Nevertheless, defining, quantifying, and assessing terrestrial C resources remain challenging, while emissions hit new records yearly. Faced with the ‘Code red’ reality for human-driven global heating [1], natural forests [2, 3], Savannas [4-6], as well as forest plantations can most easily work as carbon sinks since they fix carbon in their aboveground biomass and also store carbon in roots and soil. Due to the land use alteration, environmental services are lost. Payment for ecological benefits can be used to legitimate the option of keeping the original cover [7] or increase the recovery of degraded areas. Studies suggest that forests play a fundamental role not only in the C cycle but can also contribute to minimizing global warming due to the mitigation of greenhouse gas (GHG) emissions, including nitrous oxide (N2O), methane (CH4), and carbon dioxide (CO2) [8-10].
Therefore, removal of greenhouse gases from the atmosphere by forests and Savannas management can be considered, if not a long-term, at the very least a short-term carbon offsetting. While it won’t solve the problem alone, it will play an essential part in the emerging challenges of the Paris Agreement, such as “holding the increase in the global average temperature to well below 2°C above pre-industrial levels and pursuing efforts to limit the temperature increase to 1.5°C above pre-industrial levels” [11]
In this 21st conference of the parties (COP21), Brazil has submitted the intended nationally determined contribution (INDC) to reduce its greenhouse gas emissions in 2025 by 37%, compared with 2005 as well as cut off emissions in 2030 by 50%, compared with 2005 [12]. Regarding the land use sector, the Brazilian key policy to tackle climate change has already channeled R$ 17 billion to implement evidence-based mitigation measures within the scope of a Low Carbon Agriculture for Sustainable Development Plan (Plan ABC+). From 2020 to 2030, this public policy comprises the recovery of degraded lands; enhance nitrogen fixation, soil carbon storage, no-till farming, the integration of forest, crops and cattle breeding, agroforestry, and forest planting [12].
Brazil´s commitments also include an effort to apply at least tier 2 in methodologies for land use change, considering four C reservoirs in the forests environment: biomass (aboveground and root biomass), necromass (including dead wood and litter), and soil (soil organic material) [13]. Seen in these terms, the quality of the “Anthropogenic Emissions of Greenhouse Gases National Inventories” depends on the advancement in the use of country-specific Tier 2, which implies replacing the conservative assumptions adopted worldwide (Tier 1) with reliable data that are representative of national and regional reality.
Based on these political settings, obtaining realistic regional metrics on carbon stock measurements and GHG emissions from land use change has been a huge challenge for Brazilian scientists. Brazil has continental dimensions and a diversified economy mostly based on agribusiness products, comprising over 8.5 million square kilometers. This most biodiverse country in the entire world holds equatorial, tropical, and sub-tropical climates, as well as six distinct biomes, namely the Amazon (equatorial rainforest), the Caatinga (semi-arid), the Atlantic Forest (tropical rainforest), the Pantanal (seasonal wetlands), the Pampa (subtropical grasslands) and the Cerrado (savanna) [12].
The Cerrado biome is considered one of the world's biodiversity hotspots [14], and this region currently has the largest established agricultural area in the country, which led to the loss of 55% of the original altered area of native vegetation [15]. On the other hand, this region faces intense habitat loss [16], accounting for 26% of Brazilian emissions from land-use change [17] 10.688,73 km2 yr-1 [18].
A Review on Soil Carbon Stocks and Greenhouse Gas Mitigation of Agriculture in the Brazilian Cerrado showed a range of studies conducted on the plot scale when native vegetation in Cerrado was converted to pasture or crops [19]. However, despite the expansion of eucalyptus culture in this Savanna environment [20], scarce information is available concerning the impact of Eucalyptus plantations on carbon stocks and GEE emission [10, 21-26]. Regarding recently planted forests, Brazil added up to an area of around 10 million hectares, of which 76% were eucalyptus plantations, the world’s most planted hardwood tree [27].
Most research are restricted to aboveground biomass C and soil carbon measurements. There is a gap in estimates of the amount of C in the different Eucalyptus forest compartments, such as roots, and dead biomass in the Cerrado [25, 28, 29, 30]. As for greenhouse gas, only a few sites have been assessed in the Cerrado Biome, and the authors found that CH4 removal helped offset greenhouse gas emissions in eucalyptus stands and native Cerrado ecosystems [10, 26]. Given the importance of stands in fixing C, considering that in terrestrial ecosystems, vegetation and soil are the leading global sinks of C, information on aerial and root C is essential to reduce uncertainties in regional metrics about areas of eucalyptus stands in the Cerrado, in addition to feeding national models of C storage and climate change.
Given the hypothesis that C stocks will present different contributions between the compartments and areas studied, the following objective were formulated: Quantify, using direct methods, the biomass and C stock by compartments in eucalyptus plantations (Eucalyptus urophylla x E. grandis) of 4 and 6 years old; Estimate, by indirect methods, the biomass and C stock of the native vegetation of the Cerrado; Estimate the stock of biomass and underground C in eucalyptus stands and native vegetation of the Cerrado; Estimate C stocks in the soil up to 1.0 m depth and determine the rate of C fixation in the soil; Evaluate methane (CH4) and nitrous oxide (N2O) fluxes from the soil in eucalyptus plantations with different ages and native vegetation of the Cerrado.
- After the Introduction part, the ‘Materials and Methods’ part should come there. Without knowing the methodologies, the readers can not follow the findings of the study since these are highly technical terms
Done.
- The GWP of all GHG gases i.e., CO2, CH4, N2O and their conversions should give proper references. Are they for a 100-year lifetime basis?
Done.
Yes, the lifetime basis is 100 years. It is also mentioned in the table.
- Who produced Table 4. Is it produced from the study or from another source (Ref. is missing)?
The former Table 4, current Table 1, has listed the chemical attributes of soil samples collected specifically for the present study.
- 4.1.1-4.1.5, appropriate method (s) or allometric equations with reference (s) should be given.
1) In native vegetation, we used the indirect method of quantification of biomass which is based on the use of relationships between biomass and other variables of the tree (diameters of the base and height);
2) In the eucalyptus stands, we used the direct method, which consists of measuring the biomass through a destructive process, dispensed with allometry equations;
3) Root collections were performed in all areas by the direct method.
- Needs detail on estimation of root biomass and their amount of C (section 4.1.3).
The procedure for collecting roots from native vegetation consisted of excavating five monoliths with a circular shape, with an average diameter of 23 cm and a depth of 60 cm adapted from Gatto et al [x] method, placed in the center of each sample plot used in the floristic inventory. A total of 50 monoliths were excavated.
In collecting roots from eucalyptus stands, the collection procedure was similar to native vegetation. A medium CAP tree was selected, per plot, without flaws in the neighborhood, with a well-formed crown, which was felled to determine aerial biomass and collection of roots. Ten circular-shaped monoliths were excavated, with an average diameter of 23 cm to a depth of 60 cm, distributed in four cardinal quadrants (north, south, east, and west), based on the center stump of the eucalyptus tree felled Gatto [X].
In the tree planting line, 50 cm equidistant from the vine, two monoliths were opened, and perpendicular to the planting line (between the rows), three monoliths were opened, also 50 cm equidistant between the points from the tree vine. In separating the roots, the same procedure adopted for native vegetation was followed. The soil collected from each monolith was identified and taken to the laboratory to separate roots and other plant debris shortly after collection by manual process and with the aid of sieves.
Roots separated from the soil, and other plant materials were washed in running water to remove excess soil. Then, the root samples were weighed on a precision scale (0.01 g) to obtain the wet mass and placed to dry in a forced ventilation oven (65 ± 5 °C) until dry biomass was obtained. Subsequently, root samples collected per tree were sent to the laboratory to determine organic C using the elemental analyzer CHNS (Elementary Analyzer of the Macro Vario Cube-Elementary). After obtaining the dry mass, the roots were calculated using the volume of the cylinder: V = π.r2.h
- Section 4.1.6, line 459-467, reference mission for F value (used for the conversion from CO2 to C).
As suggested by the author, it was only considered the GWP calculations whose results are expressed in CO2 eq
- Section 4.7. Reference missing on estimation of Total carbon.
It has no reference, the equation was removed and the table of results was modified, now it shows the emission stored and the emission issued.
- Does the sign ‘Mg’ mean million grams? If it is so, then what are the differences of Mg mentioned in line 31, 125 and many other places with Mg of Table 4?
When talking about measurement units: 1 megagram [Mg] = 1 ton [t], when talking about soil attributes: magnesium is the chemical element that has the symbol Mg, which was described below the table 4, (current table 1) so there are no doubts.
- Please use CH4 instead CH4 (line 208).
Done.
- Based on the findings of AGB, Root biomass and their GWP, the authors could develop the corresponding Model (i.e., allometric equation). Then the study would be more attractive.
An interesting insisht, but our article already brings many results. Making models is interesting, but we would have to change the focus of the study.
- It is well known that half of plant biomass is generally composed of organic carbon (Table 1 and Table 2). However, the authors could simply estimate the C content from AGB, root biomass and
soil carbon, and then tabulate the GWP by using the corresponding default value of the GHGs (i.e.,CO2, CH4, N2O).
As item 7 of the review explains, we use different methods to determine biomass and carbon in native vegetation, and eucalyptus stands. In native vegetation, we used allometric equations to determine Biomass and Carbon. This is because we carried out our study in a particular area. In Brazil, according to the Forestry Code (Law 12.651/12, Article 12): Every rural property must maintain an area with native vegetation cover, as a Legal Reserve, without prejudice to the application of the rules on Permanent Preservation Areas, observing the following minimum percentages in relation to the area of the property, except for the cases provided for in article 68 of this Law: (Wording provided by Law 12.727/2012) 35% of the property located in the Cerrado area. In eucalyptus stands and for the roots of the three areas studied, the determination of biomass was made by the direct quantitative method and the determination of the Carbon content using the elemental analyzer CHNS (Elementary Analyzer of the Macro Vario Cube-Elementary). Therefore, not necessarily all compartments of the direct method gave 50% of C in the biomass; this percentage varied according to the compartment.
2)
|
|
|
N2O |
|
|
|
kg ha-1 Year -1 |
GWP1001 |
kg CO2 eq ha-1 Year -1 |
|
A1 |
0.43 (±0.23) a |
298 |
127.82 (±77) a |
|
A2 |
0.85 (±0.45) a |
298 |
271.30 (±116) a |
|
A3 |
0.33 (±0.20) a |
298 |
98.15 (±59) a |
|
CH4 |
|||
|
A1 |
-1.85 (±1.36) a |
34 |
-63.09 (±46) a |
|
A2 |
-0.98 (±0.91) a |
34 |
-33.31 (±31) a |
|
A3 |
-0.63 (±0.53) a |
34 |
-21.53 (±18) a |
1with inclusion of climate–carbon feedbacks, over a 100-year horizon. Source: Myhre [37]
- Please use the term CO2eq. emissions instead of Ceq. emissions throughout the text by calculating the appropriate value. The use of CO2eq. emissions is the most straightforward term while
describing the emissions or GWP.
Done.
- The Conclusion part also needs to be modified in accordance with the concrete findings. It is also needed to mention the limitations and applicability of this study as well as recommendations for
future research
The eucalyptus stands had a total aboveground C of 62.1 Mg ha-1 for 4-year-old eucalyptus and 81.7 Mg ha-1 for 6-year-old eucalyptus. In native vegetation, species show wide heterogeneity in terms of C storage, where Qualea grandiflora presented the highest contribution (13%) of the total 22.1 Mg ha-1 found for native vegetation. The roots had a lower contribution to C storage, ranging from 1.9 - 4.9 Mg ha-1. However, this contribution is related to a depth of 60 cm, so it is recommended for future studies that the roots be collected at least up to one meter deep. Soil C made the most significant contribution by fixing 58-84% CO2 of total C stored. In all areas, N2O and CH4 emissions were < 1 kg ha-1 yr-1. The soil acted as a continuous CH4 sink, absorbing it at rates between - 63.09 and -21.53 kg CO2 eq ha-1 yr-1. Concerning the native cerrado vegetation conversion, the “4-year-old eucalyptus stand” seemed to restore the original soil carbon stocks in the first-meter depth, regardless of some losses that might have occurred right after establishment. Conversely, a significant loss of carbon in the soil was observed due to the “13-year-long” alternative setting, where a similar natural land was converted into agriculture, mostly soybean, and then turned into the 13-year-old eucalyptus stand (18%; 28,43 Mg ha-1). Under this study, these mixed series of C baselines in landscape transitions have reflected on C dynamics outcomes, whereas at the bottom line, total C stocks were higher in the 4-year-old stand. Therefore, our finding indicates that we should consider upscaling carbon emissions and sequestration from small-scale measurements to regional scales. Moreover, it is also vitally important to attempt for uncertainties of biogeochemical cycles connected to the carbon and nitrogen cycle. Considering the CHG fluxes, we found indications of a wide range of CH4 GHG removals (12 to 50% of N2O emissions), reducing the net annual emissions of these forest systems in different proportions. For this reason, public policy approaches should rigorously quantify, besides uncertainties in metrics of low-carbon transition pathways, the ecosystem-level C stock that includes biodiversity, water cycles, and well-integrated landscape planning approaches before encouraging management practices that maximize carbon sequestration by promoting forest cover.
We would be glad to respond to any further questions and comments that you may have.
On behalf of all authors. Sincerely yours.

Reviewer 2 Report
The ms by Piontekowski Ribeiro et al. provide a study that compared carbon storage in eucalyptus stands and native Cerrado vegetation. The carbon above the ground was measured through inventory and allometric equations. Roots and soil samples were collected, and greenhouse gas emissions were measured. The results showed low emissions in all areas, with higher carbon stocks in aerial biomass for 6-year-old eucalyptus stands. However, soil carbon was lower due to management practices. The total carbon in the studied environments was able to offset greenhouse gas emissions.
The introduction provides clues to what is to follow. The ideas are good but the paragraphs need to be compressed sometimes. However, no previous studies are presented.
Then, I don't see the point of changing sections. Why are Results and Discussions presented before Methods? It makes no sense.
Although the Materials and Methods section is well structured, I had to go through almost the entire manuscript to understand both the methods applied and the study area. Until then I only suspected them. Excluding the arrangement of the sections, they are well written overall. The main purpose, however, is very hard to follow under these conditions.
Author Response
Plants (MDPI)
Dear
We thank the reviewers for their generous and thoughtful comments on our submitted manuscript. We have made all the efforts to address all the reviewers’ concerns. We have included here the reviewers’ comments and how we have addressed each point, in italics the reviewers’ comments and in color green our response.
- -Reviewer 3 –
1) The introduction provides clues to what is to follow. The ideas are good but the paragraphs need to be compressed sometimes. However, no previous studies are presented.
As the World wakes up to the need for efforts towards Climate Change mitigation strategies, the estimates of global greenhouse gases emissions and sinks have become one of the major research priorities of the scientific community. Nevertheless, defining, quantifying, and assessing terrestrial C resources remain challenging, while emissions hit new records yearly. Faced with the ‘Code red’ reality for human-driven global heating [1], natural forests [2, 3], Savannas [4-6], as well as forest plantations can most easily work as carbon sinks since they fix carbon in their aboveground biomass and also store carbon in roots and soil. Due to the land use alteration, environmental services are lost. Payment for ecological benefits can be used to legitimate the option of keeping the original cover [7] or increase the recovery of degraded areas. Studies suggest that forests play a fundamental role not only in the C cycle but can also contribute to minimizing global warming due to the mitigation of greenhouse gas (GHG) emissions, including nitrous oxide (N2O), methane (CH4), and carbon dioxide (CO2) [8-10].
Therefore, removal of greenhouse gases from the atmosphere by forests and Savannas management can be considered, if not a long-term, at the very least a short-term carbon offsetting. While it won’t solve the problem alone, it will play an essential part in the emerging challenges of the Paris Agreement, such as “holding the increase in the global average temperature to well below 2°C above pre-industrial levels and pursuing efforts to limit the temperature increase to 1.5°C above pre-industrial levels” [11]
In this 21st conference of the parties (COP21), Brazil has submitted the intended nationally determined contribution (INDC) to reduce its greenhouse gas emissions in 2025 by 37%, compared with 2005 as well as cut off emissions in 2030 by 50%, compared with 2005 [12]. Regarding the land use sector, the Brazilian key policy to tackle climate change has already channeled R$ 17 billion to implement evidence-based mitigation measures within the scope of a Low Carbon Agriculture for Sustainable Development Plan (Plan ABC+). From 2020 to 2030, this public policy comprises the recovery of degraded lands; enhance nitrogen fixation, soil carbon storage, no-till farming, the integration of forest, crops and cattle breeding, agroforestry, and forest planting [12].
Brazil´s commitments also include an effort to apply at least tier 2 in methodologies for land use change, considering four C reservoirs in the forests environment: biomass (aboveground and root biomass), necromass (including dead wood and litter), and soil (soil organic material) [13]. Seen in these terms, the quality of the “Anthropogenic Emissions of Greenhouse Gases National Inventories” depends on the advancement in the use of country-specific Tier 2, which implies replacing the conservative assumptions adopted worldwide (Tier 1) with reliable data that are representative of national and regional reality.
Based on these political settings, obtaining realistic regional metrics on carbon stock measurements and GHG emissions from land use change has been a huge challenge for Brazilian scientists. Brazil has continental dimensions and a diversified economy mostly based on agribusiness products, comprising over 8.5 million square kilometers. This most biodiverse country in the entire world holds equatorial, tropical, and sub-tropical climates, as well as six distinct biomes, namely the Amazon (equatorial rainforest), the Caatinga (semi-arid), the Atlantic Forest (tropical rainforest), the Pantanal (seasonal wetlands), the Pampa (subtropical grasslands) and the Cerrado (savanna) [12].
The Cerrado biome is considered one of the world's biodiversity hotspots [14], and this region currently has the largest established agricultural area in the country, which led to the loss of 55% of the original altered area of native vegetation [15]. On the other hand, this region faces intense habitat loss [16], accounting for 26% of Brazilian emissions from land-use change [17] 10.688,73 km2 yr-1 [18].
A Review on Soil Carbon Stocks and Greenhouse Gas Mitigation of Agriculture in the Brazilian Cerrado showed a range of studies conducted on the plot scale when native vegetation in Cerrado was converted to pasture or crops [19]. However, despite the expansion of eucalyptus culture in this Savanna environment [20], scarce information is available concerning the impact of Eucalyptus plantations on carbon stocks and GEE emission [10, 21-26]. Regarding recently planted forests, Brazil added up to an area of around 10 million hectares, of which 76% were eucalyptus plantations, the world’s most planted hardwood tree [27].
Most research are restricted to aboveground biomass C and soil carbon measurements. There is a gap in estimates of the amount of C in the different Eucalyptus forest compartments, such as roots, and dead biomass in the Cerrado [25, 28, 29, 30]. As for greenhouse gas, only a few sites have been assessed in the Cerrado Biome, and the authors found that CH4 removal helped offset greenhouse gas emissions in eucalyptus stands and native Cerrado ecosystems [10, 26]. Given the importance of stands in fixing C, considering that in terrestrial ecosystems, vegetation and soil are the leading global sinks of C, information on aerial and root C is essential to reduce uncertainties in regional metrics about areas of eucalyptus stands in the Cerrado, in addition to feeding national models of C storage and climate change.
Given the hypothesis that C stocks will present different contributions between the compartments and areas studied, the following objective were formulated: Quantify, using direct methods, the biomass and C stock by compartments in eucalyptus plantations (Eucalyptus urophylla x E. grandis) of 4 and 6 years old; Estimate, by indirect methods, the biomass and C stock of the native vegetation of the Cerrado; Estimate the stock of biomass and underground C in eucalyptus stands and native vegetation of the Cerrado; Estimate C stocks in the soil up to 1.0 m depth and determine the rate of C fixation in the soil; Evaluate methane (CH4) and nitrous oxide (N2O) fluxes from the soil in eucalyptus plantations with different ages and native vegetation of the Cerrado.
2) Then, I don't see the point of changing sections. Why are Results and Discussions presented before Methods? It makes no sense. 3)Although the Materials and Methods section is well structured, I had to go through almost the entire manuscript to understand both the methods applied and the study area. Until then I only suspected them. Excluding the arrangement of the sections, they are well written overall. The main purpose, however, is very hard to follow under these conditions.
We followed the plant journal table, but following the reviewers' suggestions, we changed the order of the sections for better understanding.
We would be glad to respond to any further questions and comments that you may have.

Round 2
Reviewer 1 Report
The authors have taken into consideration all of my comments and revised the manuscript. The new version of the manuscript looks fine to me.
Reviewer 2 Report
Indeed the ms has been much improved. It has taken into account the comments made and others I no longer have.
Ms deserves to be published in Plants.